# A Lurking Threat of Community-Acquired *Acinetobacter* Meningitis—A Rare Case Report from Punjab, India

**DOI:** 10.3390/medicines9040027

**Published:** 2022-03-31

**Authors:** Navodhya Jindal, Sonia Jain, Arghya Bhowmick, Vyom Bhargava

**Affiliations:** 1Harvard Medical Centre, Moga 142001, Punjab, India; 2Department of Paramedical Science, Lala Lajpat Rai College of Pharmacy, Moga 142001, Punjab, India; swtsoniajain@gmail.com; 3Department of Biochemistry, Bose Institute, EN Block, Sector-V, Kolkata 700091, West Bengal, India; arghyabhowmick945@gmail.com

**Keywords:** community-acquired, *Acinetobacter*, meningitis, nosocomial infection, antibiotics

## Abstract

**Background:** *Acinetobacter* spp. are a potential life-threatening cause of severe meningitis noted as a nosocomial infection after neurosurgical procedures in patients admitted to neurosurgical ICUs. Community-acquired Acinetobacter meningitis is extremely rare, and only a few cases have been reported in the literature. **Case presentation:** In this study, we report a patient from Punjab, India, who was infected after a roadside accident in which he developed CSF otorrhea and subsequent meningitis with *Acinetobacter lwoffii*. The patient was managed with the cephalosporin group of antibiotics as per the sensitivity report. For the first time, we report a rare case report of community-acquired *Acinetobacter* meningitis from Punjab, India. **Conclusions:** This case report highlights the potential pathogenicity of *Acinetobacter lwoffii* and increases concerns that this organism might rapidly evolve into a dreadful antibiotic-resistant community pathogen.

## 1. Introduction

The occurrence of community-acquired *Acinetobacter* meningitis of the central nervous system (CNS) is very rare [1,2]. *Acinetobacter* spp. are Gram-negative, opportunistic coccobacilli pertaining to the ESKAPE organisms that can accumulate diverse mechanisms of resistance, leading to the emergence of strains that are resistant to most of the commercially available antibiotics [3]. Moreover, the variable penetration of antibiotic agents through the blood–brain barrier into the cerebrospinal fluid (CSF) further limits the therapeutic choices for infections caused by *Acinetobacter* spp. [4]. Here, we present a striking case of community-acquired Acinetobacter meningitis in a patient from Moga, India, caused by the organism *Acinetobacter lwoffii*.

## 2. Case Presentation

The patient presented was brought with an alleged history of a roadside accident. The patient was hit by a car while riding on a motorcycle on 24 October 2021. The patient was found lying unconscious at the unhygienic corner of the road and was brought for emergency management. Emergency care was provided without delay. Initial examination revealed a body temperature of 38.9 °C, a blood pressure of 150/95 mm Hg, a heart rate of 110 beats/min and a respiratory rate of 16 breaths/min. An emergency computed tomographic (CT) scan of the head was conducted, which revealed a hairline fracture involving the right petrous temporal bone, as shown in Figure 1. The patient regained consciousness but was drowsy. His Glasgow Coma Score was found to be E4V4M6. The patient did not have any chronic disease and was not taking any form of medication prior to his admission. The patient did not have any known cases of diabetes, hypertension or hypothyroidism and was not addicted to any intravenous drugs. He was a smoker who smokes 1–2 cigarettes per day. Therefore, the immunological status of the patient was intact. No other comorbid history was present. The patient developed CSF leaks from both the ears after injury. The CSF leak from the right ear subsided spontaneously one day after the injury. However, the CSF leak from the left ear continued and increased initially.

## 3. Investigations

The timeline of infection aggravation and subsequent management is shown in Figure 2. On 28 October 2021, a CSF sample was taken from the left ear for a culture sensitivity test, which revealed the presence of *Acinetobacter lwoffii* sensitive to cefoperazone sulbactam, ceftriaxone sulbactam, meropenem, colistin, etc., as shown in Table 1. The antibiotic sensitivity test was conducted following the CLSI (Clinical and Laboratory Standards Institute, 2020) guidelines, an institute which is known to develop laboratory standards worldwide for medical testing [5]. Lumbar puncture was conducted the next day (29 October 2021) under all aseptic precautions, which revealed the same organism (*Acinetobacter lwoffii*) sensitive to the same group of antibiotics, confirming the presence of meningitis. The identification of the strain was further confirmed by 16S rDNA sequencing using universal sets of primers as described in previous studies [6]. The routine examination of the CSF sample showed that CSF proteins were raised to 155 mg/dL, with a normal level of glucose of 76 mg/dL. On the other hand, the CSF cell count was also raised to 1100 cells/cc, revealing neutrophilic pleocytosis consistent with the CT scan image of *Acinetobacter* meningitis, as shown in Figure 3.

## 4. Treatment

The global treatment guidelines for bacterial meningitis were followed [7]. The patient was managed with an anti-meningitic dose of cefoperazone sulbactam 3 gm IV (intravenous) every 6 h. Cefoperazone sulbactam was used as the first-choice drug in this case as it is a narrow-spectrum antibiotic which is sensitive to the organism. Hence, there would be minimal chances of developing antibiotic resistance in the organism. It was used in anti-meningitic dosage which is always higher than the normal dosage of antibiotics. The sensorium of the patient was found to be improved. The CSF leak from the left ear subsided 10 days after the injury. Additionally, the patient was managed in the right lateral position with the left ear up with head end elevation.

## 5. Outcome and Follow-Up

Lumbar puncture of the patient was conducted after 4 days (3 November 2021) of continuous IV antibiotics. The CSF culture was found to be sterile with no growth of the organism, and also the cell count and protein level returned to normal. Intravenous (IV) antibiotics were continued for 21 days. The patient became alert, conscious and mobile, indicating complete improvement. At the follow-up, 15 days after discharge, the patient was found completely fit with no side effects.

## 6. Discussion

*Acinetobacter* meningitis is a life-threatening cause of acute pyogenic meningitis which is acquired mostly after any neurosurgical procedure and is resistant to most of the antibiotics [1,2]. Community-acquired cases of *Acinetobacter* meningitis are generally sensitive to most of the empirical antibiotics [8,9]. In this rare case, a patient from India developed *Acinetobacter* meningitis after trauma where no neurosurgical procedure was performed. To date, very few cases of *Acinetobacter* meningitis have been reported, and most are nosocomial infections which are acquired in the hospital ICU environment and are highly resistant to most of the commercially available antibiotics, leaving little choice of antibiotics to treat the infection [10,11]. This observed report of *Acinetobacter* meningitis by *Acinetobacter lwoffii* is a clear indication that the threat has moved far beyond the hospitals in India. *Acinetobacter lwoffii* may soon develop into a significant dreadful community pathogen in India.

## 7. Conclusions

Hospital-acquired *Acinetobacter* meningitis can be community-acquired too. The unscrupulous use of antibiotics should be avoided to prevent the development of highly resistant strains of *Acinetobacter* spp. so that all the narrow-spectrum and commercially available antibiotics can be targeted against them.

## Figures and Tables

**Figure 1 medicines-09-00027-f001:**
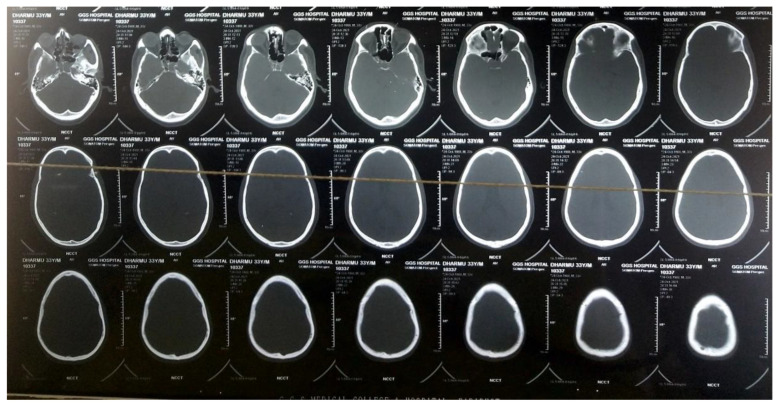
CT scan image showing hairline fracture of the right petrous temporal bone of the patient.

**Figure 2 medicines-09-00027-f002:**
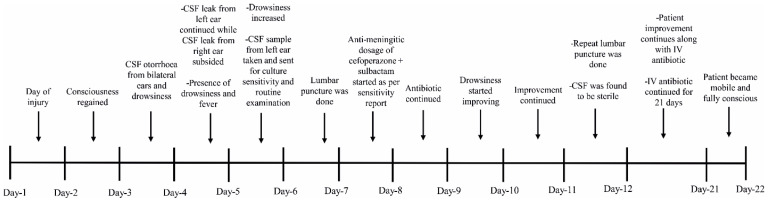
Timeline of disease aggravation of the affected patient and subsequent management of the disease by proper antibiotic treatment. (CSF—Cerebrospinal Fluid, IV—Intravenous).

**Figure 3 medicines-09-00027-f003:**
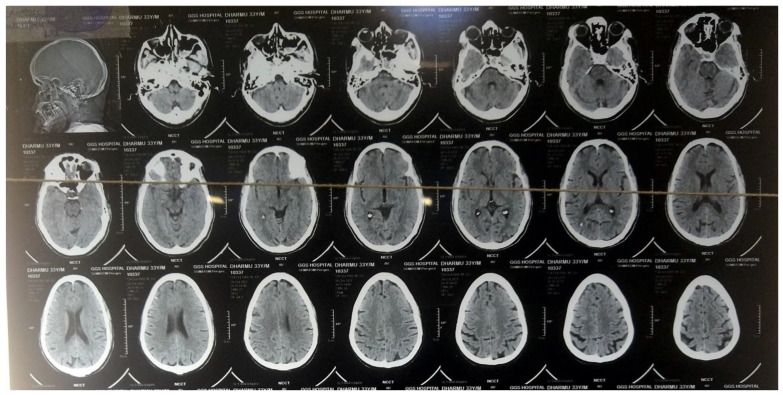
Computed tomographic (CT) scan image showing diffuse hypodensities consistent with meningitis.

**Table 1 medicines-09-00027-t001:** Antibiogram profiling of *Acinetobacter lwoffi* showing sensitivity to various antibiotics (R—resistant; S—sensitive; IM—intermediate).

Antimicrobials	Interpretation
Gentamycin	R
Cephalaxin	R
Amikacin	R
Ciprofloxacin	R
Netilmycin	R
Cefotaxime	R
Ceftazidime + Sulbactam	R
Ofloxacin	IM
Cefoperazone + Sulbactam	S
Augmantin	R
Cotrimoxazole	R
Meropenem	S
Pipercillin/Tazobactum	S
Gatifloxacin	S
Cefuroxime	R
Cefixime	R
Cefepime	R
Cefprozil	R
Ceftrixone + Sulbactam	S
Imipenem	IM
Colistin	S
Ertapenem	R
Elores	S
Levofloxacin	S

## Data Availability

All the relevant data pertaining to the patient are within the paper. Further inquiries can be directed to the corresponding authors.

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
