# Peer review of "A Lurking Threat of Community-Acquired Acinetobacter Meningitis—A Rare Case Report from Punjab, India"

_medicines, 2022, doi:10.3390/medicines9040027_

Round 1

Reviewer 1 Report

This paper is a case report on a successful case of treatment for meningitis patients by CA-Acinetobacter Lwoffii, and it seems to contain medically important information. The paper is well written, but it seems that some corrections are necessary. 

  • Reference1 and 2 were not within the last 5 years. Please quote the latest references.
  • If there is a basic disease of the case and a drug being used, it should be described.
  • Is the detected bacteria a metal-β lactamase-producing bacteria? Is it not inspected?
  • Author should describe why cefoperazone plus sulbactam was used in the treatment as a first-choice drug.
  • Author should describe the examination findings at the time of hospitalization. Especially, the immune function of the case is unknown.
  • It is necessary to refer the grobal treatment guidelines for the bacterial meningitis.

Author Response

  • 1) Reference1 and 2 were not within the last 5 years. Please quote the latest references.
  • Response 1- Thank you for your kind suggestion. We have modified the references accordingly in the final manuscript.
  • 2) If there is a basic disease of the case and a drug being used, it should be described.
  • Response 2- Thank you for your comment. The patient did not  have any known cases of chronic disease and was not taking any medicines before the accident. This piece of information has been added to the manuscript.
  • 3) Is the detected bacteria a metal-β lactamase-producing bacteria? Is it not inspected?
  • Response 3- Acinetobacter lwoffii detected in our case was not a metal-beta lactamase producing bacteria as it is sensitive to almost all the antibiotics including the narrow spectrum antibiotics (antibiogram profile data). So it was not detected.
  • 4) Author should describe why cefoperazone plus sulbactam was used in the treatment as a first-choice drug.
  • Response 4- Thank you for your kind suggestion. The description has been added to the final manuscript. Cefoperazone sulbactum was used as first- choice drug in this case as it is a narrow spectrum antibiotic which was sensitive to the organism. Hence, there would be minimal chances of developing antibiotic resistance among the organisms  as per the antibiotic protocol. It was used in anti-meningitic dosage which is always higher than normal dosage of antibiotics. 
  • 5) Author should describe the examination findings at the time of hospitalization. Especially, the immune function of the case is unknown.
  • Response 5-  Patient did not  have any known cases of diabetes, hypertension or hypothyroidism. He was not addicted to any intravenous drugs. He was a smoker but smokes 1-2 cigarettes per day. So immunological status of the patient was intact. This piece of information has been added to the manuscript.
  • 6) It is necessary to refer the grobal treatment guidelines for the bacterial meningitis.
  • Response 6- Thank you for your kind suggestion. The reference to global treatment guidelines fot the bacterial meningitis has been added to the final manuscript.

Reviewer 2 Report

The manuscript contains some very interesting results.

The manuscript described a case report of a patient from Punjab, India, who developed CSF otorrhea and subsequent meningitis with Acinetobacter lwoffii after a roadside accident. The manuscript contains some very interesting results and is suitable for publication after minor revisions.

 Minor revisions

Standardize Acinetobacter lwoffii and Acinetobacter throughout the text.

Lines 23-24: The sentence should not be in italics;

Line 24: The is not in italic;

Line 29-30: community acquired is not in italic;

Line 31: Acinetobacter lwoffii is in italic;

Line 38: In the first time, describe in full what CT means;

Line 61 and Line 82: There is only one table in the text, so it must be Table 1 and not Table 3;

Lines 62-63: Describe what CLSI means and after placing the acronym in parentheses

Line 64: lwoffii starts with a lowercase letter;

Line 82: Table 3 must be improved;

Line 70 and Line 99: There is no figure 3, so the numbering of figure 4 must be revised;

Lines 140-163: Review all references, especially capitalized ones (references 3, 4, 6, 9 and 10).

Author Response

1)Standardize Acinetobacter lwoffii and Acinetobacter throughout the text.

Response 1- We apologize for the inconvenience caused. This has been standardized throughout the text.

2) Lines 23-24: The sentence should not be in italics;

Response 2- Thank you for your comment. The sentence style has been changed from italics to normal font. 

3) Line 29-30: community acquired is not in italic;

Response 3- Thank you for your comment. The community acquired has been changed from italics to normal font. 

4) Line 31: Acinetobacter lwoffii is in italic;

Response 4- Thank you for your comment. Acinetobacter lwoffii  has been changed to italics. 

5) Line 38: In the first time, describe in full what CT means;

Response 5- Thank you for your valuable suggestion. The description of CT in full has been added to the final manuscript.

6) Line 61 and Line 82: There is only one table in the text, so it must be Table 1 and not Table 3;

Response 6- Thank you for your kind suggestion. The table annotation has been changed accordingly in the final manuscript.

7) Lines 62-63: Describe what CLSI means and after placing the acronym in parentheses

Response 7- Thank you for your suggestion. The meaning of CLSI has been added to the final manuscript.

8) Line 64: lwoffii starts with a lowercase letter;

Response 7- Thank you for your comment. This has been changed in the final manuscript.

9) Line 82: Table 3 must be improved;

Response 9- Thank you for your comment. The table 3 has been improved in the final manuscript.

10) Line 70 and Line 99: There is no figure 3, so the numbering of figure 4 must be revised;

Response 10- Thank you for your comment. The numbering has been revised in the final manuscript.

11) Lines 140-163: Review all references, especially capitalized ones (references 3, 4, 6, 9 and 10)

Response 10- Thank you for your valuable comment. The references has been revised accordingly in the final manuscript.
